# Hard to Reach and Hidden: Improving the Identification of Young Dementia Carers

**DOI:** 10.3390/ijerph20237103

**Published:** 2023-11-23

**Authors:** Patricia Masterson-Algar, Kieren Egan, Greg Flynn, Gwenllian Hughes, Aimee Spector, Joshua Stott, Gill Windle

**Affiliations:** 1School of Medical and Health Sciences, Bangor University, Bangor LL57 2EF, UK; g.flynn@bangor.ac.uk (G.F.); seub04@bangor.ac.uk (G.H.); g.windle@bangor.ac.uk (G.W.); 2Department of Computer and Information Science, University of Strathclyde, Glasgow G11 XH, UK; kieren.egan@strath.ac.uk; 3Department of Clinical, Educational and Health Psychology, University College London, London WC1E 6BT, UK; a.spector@ucl.ac.uk (A.S.); j.stott@ucl.ac.uk (J.S.)

**Keywords:** young person, young carer, adolescents, younger onset dementia, dementia, carer, admiral nurse, schools

## Abstract

Young dementia carers (YDCs) rarely receive appropriate training and support. Their visibility and identification remain dangerously low, and, consequently, support initiatives being developed are failing to reach them. This study explored the success (or failure) of YDC identification pathways as well as the barriers and enablers to their implementation. An explorative qualitative approach was followed, drawing on the experiences of parents of YDCs, dementia researchers, professionals in the field of dementia/young carers, and young adult carers. Data collection involved semi-structured interviews (*n* = 17) and a participatory 2-h workshop to discuss and critique preliminary themes as well as explore strategies to increase the visibility and identification of YDCs. Five themes were identified: a “whole-family approach” (as a pathway to identification), “not a carer” (self/family identification), a postcode lottery (high variability of support services), tailored support that is “fit for purpose”, and the “power” of peer support. Recommendations on potential initiatives and actions that can help raise awareness and increase the identification success of YDCs are proposed. Our findings support the need for a broad and holistic approach to the identification of YDCs that runs alongside the development of support initiatives that are accessible and relatable. The support itself will play a role in improving subsequent identification or hindering it if not “fit for purpose”.

## 1. Introduction

The World Health Organization [WHO] [1] has declared dementia as one of the greatest healthcare challenges of the 21st century. In the UK, approximately 885,000 people are living with this condition. This is equal to 1 in every 79 (1.3%) of the entire UK population and 1 in every 14 of the population aged 65 years and over. The total number is predicted to increase to over 1.6 million by 2040 [2].

Across the world, the care and support for people living with dementia are primarily provided by family members [3,4]. They assist in all aspects of daily life, including personal care, but vitally, they also play a major role in supporting the individual’s mental wellbeing and sense of belonging [1,4]. Although spouses and adult children are, in most cases, the primary caregivers, often younger members of the family take on an active role in supporting their family member with dementia [5]. The number of children in this situation is rising. This is due to more families taking on the care of a grandparent [6,7] and people with young onset dementia (YOD) being more likely to have school-age children, particularly as couples are waiting longer to have children. These young people can be classed as “young carers” as they are under the age of 18 and undertake unpaid care [8]. Although positive outcomes such as improved self-esteem and resilience have been associated with the young carer role [6], evidence also shows a negative impact on personal relationships and overall wellbeing as a result of increased anxiety and levels of stress, loneliness, and isolation [9,10]. More specifically, accumulating evidence highlights the impact that parental YOD can have on these children as they experience the gradual loss of the parent at an age when they are developing their own identity [11,12,13]. This diagnosis has an impact on family roles and future plans [14], and as the condition progresses, young members may take on more caring responsibilities [15]. Similarly, a recent review by Venters & Jones [7] has described positive and negative impacts of grandparental dementia on young people. Regardless of this evidence, and despite their statutory rights [16], young carers in the UK (but also worldwide) are under-supported [8], and this reality is exacerbated in the case of young dementia carers (YDCs) who are vastly overlooked [11,13]. 

Evidence shows that identifying YDCs is challenging. Hence, there is a strong need for raising awareness regarding their needs [11,13,17]. In 2016, the National Children’s Bureau [18] undertook research in England aimed at collating evidence on how the needs of YDCs could best be met. Online survey data were collected from practitioners and managers working in services for young carers, dementia carers, and/or carers more generally. Despite extensive dissemination efforts, the authors reported significant issues with data collection (only 12 respondents completed the survey) and concluded that there is a need to address the invisibility and “hidden nature” of this group of young people who help care for an adult with dementia. More recently, work carried out by The Children’s Society [19] reported similar concerns, identifying YDCs as a “seldom heard” group with worryingly low levels of self-recognition associated, in part, with the social stigma linked to having a parent with dementia. 

In response to the above, tailored support tools are being developed. Very recently, Masterson-Algar et al. [20] developed iSupport for Young People. This is an adaptation of iSupport, an e-health training program developed by the WHO to support adult dementia carers [21]. iSupport for Young People aims to support the mental health, knowledge, and skills of YDCs and help service providers improve the support they offer. However, as previously explained, the visibility and level of identification of these young people remain very low, and so the worry is that tools such as iSupport for Young People will not reach their target audience, therefore failing to help mitigate the impact of caring and support these young people and their families. In light of this, the aim of this work is to explore the factors that play a role in the success (or failure) of relevant services/organizations and research to identify (and support) YDCs. Specifically, we explore:Current pathways for identification of YDCs and barriers and facilitators to their implementation.Potential strategies that can improve the visibility of YDCs and, in turn, raise awareness.

## 2. Materials and Methods

A qualitative approach was followed, drawing from the lived experiences of study participants. Recruitment took place during February and March 2023, and purposive sampling based on professional background and lived experience was used. Team members’ networks and relevant social media platforms were used to approach potential participants. To be eligible, participants had to belong to one of four categories: a parent of a YDC; a young adult (16+) with experience in caring for a family member with dementia at a younger age; a dementia researcher; and a professional working in the dementia/young carers sector. This could include local authority, publicly and privately funded organizations, as well as charities. 

Ethical approval for this study was obtained from Bangor University’s School of Medical and Health Sciences Academic Ethics Committee (ref: 2021-16915-A14954) on 19/01/2023. Those who expressed an interest in taking part were provided with a participant information sheet, given the opportunity to ask questions, and asked to provide written consent.

### 2.1. Data Collection

All data collection was conducted online and involved two phases: 

Phase 1: Semi-structured interviews: Interviews were carried out by PMA, who arranged to meet the participants on Zoom. The interview spine (see Appendix A) was agreed among all team members and was specifically designed to capture information on personal experiences and awareness of the visibility and identification pathways of YDCs. Interviews were expected to last approximately one hour and were recorded and transcribed verbatim. 

Phase 2: Participatory online workshop: once all interviews were completed, participants took part in a 2-h online workshop. They were involved as “knowing subjects” and asked to bring their perspectives into the knowledge-production process [10]. During the first part of the workshop, participants had the opportunity to discuss and critique results from the preliminary analysis of phase 1 data (emerging themes) that PMA presented (see Appendix A). The second part focused on exploring strategies to increase the visibility and improve the identification pathways of YDCs. The workshop was audio recorded and transcribed verbatim.

### 2.2. Qualitative Analysis

An iterative process of explanation building and detailed inductive thematic analysis was carried out following the six steps of the method described by Braun and Clarke [22]. The first author (PMA) read all transcripts several times, searching for patterns and meanings. Initial coding was carried out using Atlas.ti. Preliminary themes were generated and shared with the research team for comments, and when necessary, agreement was reached through discussion. Workshop data were subsequently coded, and with input from all authors, preliminary themes were refined following an iterative process of explanation building. As a result, final themes addressing study objectives were generated. 

## 3. Results

A total of 23 potential participants were contacted through team members’ networks. No participants were recruited via social media. Four of them declined due to work and personal commitments. The remaining 19 consented to take part in the study (see Table 1 for details). Eleven participants were professionals working in a range of relevant settings across the UK, such as dementia/young carers charities, local councils, and health and social care. Three were experienced dementia researchers, and the remaining five participants had lived experience of dementia and YOD within their family. Four of those were parents to a YDC, and one was a young adult carer. Seventeen participants completed the online interview, which, on average, lasted between 40 and 60 min. Fifteen participants attended the online workshop, which was 2 h long (including a comfort break). 

### 3.1. Identified Themes

Interview data analysis identified seven preliminary themes. These were further refined in response to participants’ input during the phase 2 workshop and team discussions, leading to five themes (and sub-themes) (Table 2). The first three directly relate to the study’s aim, looking at current pathways of identification of YDCs. Themes 4 and 5 discuss strategies around the design and content of support initiatives, and they address the study’s aim of identifying strategies that can improve visibility and, in turn, raise awareness of YDCs. Themes will now be described with exemplary evidence from transcripts.

#### 3.1.1. Theme 1—A “Whole-Family Approach” [as a Clear Pathway to Identification]

There was consensus across all participants on the vital importance of a family-centered approach to supporting those affected by dementia and the role that this approach can play in identifying YDCs. Unanimously, everyone agreed that dementia shapes the lives of all family members and that often, particularly in the case of YOD, children under the age of 18 must deal first-hand with its multi-layered impact. A family approach was described as vital to “untangle” the impact of the illness. As one participant put it:

“*When there’s dependents, you know, children under 18 involved. It needs to be a family-centered approach to the dementia care. It isn’t person centered approach. They are a unit, and all the care and support needs to be for the whole-family unit because they all need it.*”(P15)

Participants described several barriers to the implementation of family-centered approaches, which in turn had a negative impact on YDCs’ identification. Current processes in place when adults are newly diagnosed with dementia do not require professionals to enquire about younger family members who are likely to be affected. These post-diagnosis conversations often remain solely focused on identifying the primary caregiver and almost never explore the role and potential impact on younger family members. One participant explained:

“*If their systems and processes don’t ask around are there any children who might be affected…Then I think that’s missing a trick and I think that’s one thing, if we really want to improve identification of this group, then that’s one of the real opportunities.*”(P6)

If the post-diagnosis process does not include exploring the wider family members of the person with dementia, identification of YDCs then comes down to individuals “thinking outside the box” driven by their own interests. As the same participant continued to explain:

“*Particularly within healthcare settings, if it’s not part of the process, in terms of asking a question then it very much relies on that individual having that knowledge. Because that’s a prompt, that’s what that instigates, because they will be brilliant people who get it, who will do it anyway. But then it’ll be for those who are sort of not so familiar, to ask the question and think about it.*”(P6)

Leading from this, participants also explained that in their view, identification was also reliant on not only upskilling professionals (primarily those working within adult services and settings, including dementia-related charities) to make sure they understand what a family-centered approach is but also making sure they are aware of all facets of caring. This means moving away from the reductionist view that defines it as physical/personal care provided by the primary carer:

“*A lot of work on identification focuses on those who provide physical care for someone rather than the emotional support, so say if it’s a grandparent with dementia, they [YDCs] may well be providing emotional support to the parent, who’s dealing with that side of it as well. So, identification within adult settings is really, really key.*”(P6)

Identification of YDCs via adult services and adult support initiatives was, in everyone’s view, critical. However, Adult Social Care settings were described as “not historically great at identifying young carers”, and this was reflected, as one participant explained, in a very small number of referrals from these services to young carers organizations. A lack of communication and confusion in terms of available referral pathways was identified as one of the primary causes that leads to these young people being left behind. One participant explained:

“*There was also a misunderstanding as to who would refer the young person. So, education presumed social work would be referring and social work presumed education staff would be referring. And it turned out no one was referring the young person, so they were just kind of left dangling, which is really sad.*”(P1)

Other examples of adult support settings that were mentioned could be used to identify YDCs were care home providers and charity-run dementia support groups for people with dementia and their carers:

“*Visiting dementia support groups and speaking to the carers, and seeing if they’ve got children, seeing if they’ve even got grandchildren who are there and who are helping to support (the person with dementia).*”(P1)

#### 3.1.2. Theme 2—“Not a Carer”

Participants, particularly the parents of YDCs, reported difficulty in viewing their children as “young carers”. Parents described different reasons for this. One participant explained:

“*I didn’t see my sons as carers, because they’ve always looked after Nana and Grandpa. When they went out, they’d always made sure they were OK. So, as dad’s dementia progressed the way they sort of cared for him just changed. Until I actually sat down and thought, well actually yes, they are. It was just something that they adapted into live. That’s how they lived.*”(P8)

In line with what was described in Theme 1, another reason parents did not see their children as “carers” was due to them associating the role of carers as purely linked to day-to-day self-care tasks. As one participant explained:

“*They weren’t carers because they didn’t chop up his food or look after him. But yeah, I was at work. And who was babysitting who? […] It’s just that these kids get a higher degree of responsibility at an earlier age. I wouldn’t have called them young carers, but yeah, they were. They’re babysitting their dad at, you know, at age 7, 8, 9 so I can go to work.*”(P15)

Professional participants and researchers agreed with the above. The label “young carer” was described as leading to confusion. It was perceived as reductionist if it ignored all aspects of emotional support, which are often what YDCs are mostly involved in (on a regular basis). This label was described as generating a “pushback” that “puts people off”. As one participant described:

“*We use that term quite loosely and broadly, you’re the carer. And there’s often a pushback on that, from loads of people I’ve worked with. No actually, I’m a wife to my husband, or we’re friends, or I’m a child, you know, this is my mum, I’m not my mum’s carer, I’m my mum’s daughter.*”(P4)

Similarly, according to our data, YDCs themselves do not tend to identify as carers. They perceive caring as a “normal part of their life”, and many of them, who were very young at the time of their family member’s diagnosis, do not remember any different. Also, as these young people are often not the primary carers, their responsibilities are occasional. As one parent explained, “*they dip in and out as required rather than it being ‘Oh, I have to get up and I have to help him get washed. I have to help him get dressed.’*” (P7).

A further reason that participants explained for YDCs avoiding being identified as “young carers” is linked to the perceived negative connotations that this could have in their day-to-day life at a time when they are developing their own sense of self and identity:

“*People don’t understand it. Someone with young onset might look perfectly normal on the outside so kids can become very secretive. They can become very isolated because they don’t want friends to come home and see what’s going on. All that emotion and turmoil and trying to work out what the hell is going on.*”(P5)

The difficulties and reticence of young people to self-identify as a young carer were also described as closely linked to the way in which their parent’s YOD diagnosis was explained to them by their “healthy” parent. Participants explained that although not disclosing information is often seen by parents as a way to protect the child from the “unknown”, it can lead to young people feeling anxious and confused. As two parent participants explained:

“*I find that the more open parents are with their children at an earlier stage, the more they are able to come to terms with what’s going on.*”(P5)

“*It wasn’t a big elephant in the room. It wasn’t a big dark secret. It wasn’t. It was talked about. So therefore, when it got worse, I could say, look, I’m really sorry, girls, but dad’s dementia is getting worse. And this is why we now can’t do this, or this is why he finds this difficult. At each stage, you could explain it.*”(P15)

#### 3.1.3. Theme 3—Postcode Lottery

Participants voiced their concerns about the high variability of support services for YDCs provided by schools, local authorities, and charities across the UK. This was seen as having a direct impact on levels of identification. Areas with well-established support services (in which some of the participants lived/practiced) had, over the years, developed and implemented several initiatives to support YDCs and their families. An example discussed in detail was the role of Admiral Nurses (specialist dementia nurses) who provide support to families affected by dementia. Although their support was unanimously described as “gold standard” by all, Admiral Nurses are not available across all the UK nations, which resulted in frustration across all participant groups. As one participant explained:

“*I had been living with my husband’s dementia for about 10 years and somebody said to me, oh, have you not got any? Well, there were no Admiral Nurses in [city]. None at all. I think these days most people who need a Macmillan nurse can get access to one. But you know it shouldn’t be like that.*”(P12)

Admiral Nurses who took part in this study described how, in line with the nature of YOD, their role is to provide support for the families as and when needed and that, not surprisingly, it is often emotionally draining.

“*I love the families I work with. I’m not going to lie; I do struggle with the emotional overload sometimes. You just feel so frustrated, you know there are no magic wands. It is that ability as Admiral Nurses for families to feel held. And not feel completely on their own. They know they can rely on you being there for them at a time when they just need to be heard. And not having to tell their story again and again, not having to explain themselves.*”(P5)

The same “postcode lottery” aspect applied to the role that schools across the UK can offer in terms of helping improve the identification and support available to YDCs. Unanimously, participants agreed that schools should play a vital role in identifying and supporting these young people. However, because of the pressures faced by schools, according to participants, this role is, in the best-case scenario, limited to that of a “port of call” and signposting:

“*We are not asking them (schools) to do everything, it’s about them being the ones to identify potential children in need of support. And then linking them into the relevant organization, whether it be through the local authority, whether it be a carer’s organization or a dementia specific organization, each local area will need their own pathway for it.*”(P6)

Participants agreed that across the UK, schools show high variability in terms of how effective and invested they are in identifying and supporting YDCs. Personal experience of dementia of relevant school staff (e.g., headmasters, pastoral care team) was described as one of the factors contributing to this variability:

“*If there is someone within the school who has had their own personal experience of dementia that opens doors. My girls were at primary school, and having the head teacher, who had an experience of having a mother with dementia, meant she was much more sympathetic to me when I burst into her office and went blah, help!*”(P15)

The difference was reported to be more accentuated between primary and secondary schools due mainly to size. Professional participants also conveyed difficulties engaging, particularly secondary schools, in their research projects or support initiatives.

“*At primary school, they were very well supported, and actually, the school invited me and her to go and do a talk for the teachers, about dementia, and what it’s like to care for somebody in your family with dementia, and I thought that was absolutely fantastic that they wanted to know how to support her. At secondary school, where nobody even asked, it’s almost like they really didn’t want to know, because it might be too difficult.*”(P2)

A lack of clear referral pathways that schools can follow, as well as a lack of awareness and understanding of what a “young carer” was, was unanimously described as having a major impact on the ability of schools to identify and support these children.

“*I surveyed all of our education staff, primary and secondary school. And I had almost 800 responses, so we only have about 1200 staff, so it was a really great response rate. And almost 90% of those staff have never had any training whatsoever on young carers.*”(P1)

“*We don’t have any set standard referral pathways for children and families living with young onset dementia. I think they’ve recently been described as the kind of forgotten children.*”(P4)

#### 3.1.4. Theme 4—Tailored Support That Is “Fit for Purpose”

The uncertain and unpredictable nature of dementia, and YOD in particular, was described by most participants as an added factor that brings unique challenges for families dealing with the diagnosis. Hence, parent participants conveyed how, in their opinion, there is a lack of support within the young carers charity sector that is “fit for purpose”:

“*I’ve never found that right place for them. They were seen as young carers, but what they were offered were things like trips to bowling or to a trampoline park, and although of course they love all of that sort of thing, I can take them there, and the young carers that they were with absolutely need that respite. My girls didn’t particularly need that, they needed someone to talk to.*”(P2)

However, there was agreement among participants on the fact that recently, some charities that support young carers have started to realize this reality and are actively moving towards a more young-person-centered approach:

“*What should young carers’ support look like? I think there are lots of young carers for whom the trips, the breaks are a hugely important part. But what else does the Young Carer Service do, how does it achieve some of the long-lasting change for that young person? What are the biggest issues for them? And I think that’s where lots of Young Carer Services have moved to, that person centered way. Is it trips and activities or is it actually that information about the condition? Is it someone to talk to? Is it about the emergency planning in place? I think it’s having that variety of support.*”(P6)

YDCs, who often do not see themselves as carers, as previously discussed, are not likely to engage directly with these charities and respond to their marketing campaigns. This means that the vital role they should play in identifying and supporting these young people is lost. As one professional explains, in response to this, targeting parents is the strategy used in some of these charities:

“*It’s also whether actually will young people access. If you are a young carer’s charity, will they be following you on social media? Probably not. If you’re a young people’s charity, possibly might be, if they’re interested in mental health, but otherwise how do you get that message to the young people? I think a lot of the focus is around getting the messaging to parents, and professionals, because are they going to see it? And then if they do see it, what can they do, how can they access it?*”(P6)

Support for YDCs should be offered from diagnosis and continue throughout its progression. Participants, Admiral Nurses in particular, explained that it is often the case that in the early stages of dementia, young people (and their family members) are not ready to ask for help and receive support. However, as they discussed, it is about them knowing that the support is there whenever it is needed and understanding there is a period of adjustment:

“*Once the family has that diagnosis, there is that period of adjustment, it’s life changing, and the family needs to work through together, and you don’t want to be kind of running in and saying, you know, this service is available. But it’s about saying, this is the diagnosis, this is what it means. And I’m here for you and I’m going to come back.*”(P4)

“*At the point of diagnosis, find a way to get in touch with them, and have something useful for them. If they’re not ready for it then, make sure that it’s followed up. So that when they are ready, you’re ready to go. There needs to be that iterative process involved. Just so that people know that when they do need help, that it’s there.*”(P7)

One parent who never had access to Admiral Nurses explained it in this way:

“*We didn’t need a lot of support; we went along quite nicely most of the time. But there were those times, okay things have changed, or this is a different dynamic now, or GCSEs and this… And it would have been so helpful if I’d just been able to pick up my phone to the one person and say could you help out with this, I’m not quite sure what I’m doing as a mum, or as a carer, right now.*”(P2)

Overall, participants agreed that it is often at crisis point that young people finally reach out for support. This was linked not only to the progression of the condition (e.g., hospitalizations, transition to domiciliary care) but also to their own development as adolescents who are emotionally growing:

“*Or just immaturity, I think everybody thinks that they can do it themselves, even in advanced years, you struggle on and do it, and eventually you think actually, you know what, I can’t do this on my own.*”(P7)

Finally, linked to the need to tailor support services, participants discussed at length the nuances between those young people affected by either parental or grandparental dementia. Overall, there was agreement on the fact that the impact of YOD is multi-layered and unique, as it is the young person’s primary caregiver (parent) that is “lost”. Hence, although services should aim to support all young people affected by dementia, there is a strong and urgent need, conveyed by participants, to offer tailored services to address the needs of this cohort of children affected by YOD:

“*If you are a 3-year-old when your parent gets dementia, you’ll never remember that parent without dementia. And it’s trying to get people to understand. Why this cohort of children and young people’s needs are different. It is really crucial in terms of developing services to support them.*”(P5)

#### 3.1.5. Theme 5—The “Power” of Peer Support

Unanimously, all participants agreed that peer support was an effective way of supporting YDCs to deal with the impact of the illness at its different stages:

“*Talk, communication. This should be the what the memory services should be teaching. That the way your kids are going to get through it is by peer support and knowing they’re not alone. And by talking about it and talking to their parents. That’s what is going to be a crucial factor.*”(P15)

Peer support was described as a powerful tool that generates feelings of “camaraderie” where everybody is aware, without having to talk about it, of the challenges they and others are facing (e.g., feeling left out, isolated, misunderstood). As two participants explained:

“*Our children felt unusual, isolated and alone. We organized art therapy workshops, but it could have been anything. I don’t think they specifically spoke about the fact that they all had a parent with dementia. It was that camaraderie without it being said out loud. I might be the only one in my school but there are other kids who are going through what I’m going through. That socialization was important to them.*”(P15)

“*In that group there’s a universal understanding, and I don’t then have to do the explaining part. And it’s the explaining part that’s the friction in society for myself. But in this group, we are all a part of this, we’re doing it together, not alone, and we don’t have to really say as much or as little.*”(P17)

The success of peer support also lies, according to participants, in the fact that it generates a feeling of normality:

“*Just bringing young people together, of different age groups, but not necessarily talking about dementia, the therapy was actually just being with other people and just doing normal things. They don’t want to be singled out, they don’t bring their friends home, or want to have to explain themselves. So being with other people who are in a similar situation, is really, really powerful.*”(P5)

Finally, in line with Theme 1, participants reinforced the fact that this peer support works best sometimes if targeting the family as a whole:

“*When families come together, they do gravitate towards another family who experiences the same things just because they have lived experience and they can share those things and they form really good relationships outside of the group. And they talk and it’s providing them an opportunity to expand their networks and share, and that important.*”(P10)

## 4. Discussion

The needs of YDCs (and their families) are not being met. Our findings explored in detail the multifaceted factors that play a role in enabling or hindering identification and support efforts. Not surprisingly, our themes present clear parallels with published evidence focusing on young carers in general [23] and on adult carers [24,25,26]. For example, a recent study that surveyed almost 2500 young carers across six European countries, including the UK, described service provision as disjointed and lacking in continuity [27]. This had a direct impact on the proportion of young carers receiving formal support, which, for example, was reported to be as low as 13% and 15% of young carers in Italy and Switzerland, respectively. Data from the UK showed that 47% of young carers had received formal support; however, the authors attribute this higher number to the way in which young carers in the UK were recruited, which was via young carers’ projects (rather than via schools). Similar evidence has been reported on adult carers. Carers UK [24] data showed that over a third of adult carers (36%) did not know what services were available to them, and 61% were worried about what services would continue for the next 12 months. The data also showed that only 25% of carers had completed a Carer’s Assessment, and, most worryingly, 39% of those who had not completed one did not know what they were. According to our evidence, these same issues apply to YDCs who, depending on where they live, might have access to some type of formal support but, even then, are likely not to access it because they are not aware of it or because they do not see themselves as the target audience. The caring role that young people play, not only in the UK but also internationally, remains a “*private family matter rather than an issue for public policy intervention*” [8] (p. 82). In the UK, this is despite campaigns such as “Young Carer’s Awareness Day” organized by the Carers Trust since 2014 and new legislation protecting their rights [16,17,28].

Our findings provide evidence to suggest that low level of self-identification is likely to be more prevalent among YDCs as well as among other young carers who are affected by parental conditions that are particularly stigmatized, such as those linked to alcohol/substance misuse, mental health, or Huntington’s Disease [19,23,29]. We know that children and young people in these families often do not see themselves as carers but also that they actively avoid being identified as one for fear of discrimination and prejudice in school or among friends [9,10]. As Astrup [30] reported, this fear is underpinned by a lack of understanding of what the term “carer” means; this is particularly true for those young carers who do not receive any formal support [23]. Hence, we believe that it is by working with young people themselves and relevant stakeholders to create services that are “fit for purpose” that some of the challenges of self-identification can be overcome. As we see it, without tailored services in place, identification success becomes meaningless, even harmful, potentially driving YDCs and their families away.

According to our participants, adult services are failing at identifying YDCs (and young carers in general). Our data are in line with previous research that showed how, across Europe, only 4–10% of referrals to young carer services were done by adult social care and health services [27]. To address this, in recent years, adult services have been pressured to make a “paradigm shift” and move towards a whole-family approach. However, evidence (including this study) suggests that this approach is currently not being successfully implemented in the UK [8,19,25] or across Europe [23,31]. This failure is linked to communication gaps between children’s services and adult services, coupled with a lack of awareness among local authorities and health professionals on what a young carer is and what are the referral pathways [32]. Although there is mounting pressure on social services and the NHS to upskill professionals to “*ensure they are able to identify, signpost and support carers when they encounter them*” [24] (p. 10), in our opinion there is a strong need for more conceptualization and more professional awareness in both social and health sectors of whole-family practices with emphasis on the need to move away from a short-sighted definition of “family” as a dyad formed of the primary carer and the person being cared for [32,33,34]. This should lead to improvements in the way each family member’s needs are listened to and addressed, not only in dementia care but also in the care of families affected by Parkinson’s [35] or stroke [36], where a lack of family-centered care has also been identified.

Our findings advocate for services like the ones Admiral Nurses provide in the UK as key drivers to the successful implementation of a whole-family approach and the identification of YDCs. These services are provided by registered nurses who, supported by the charity Dementia UK, specialize in supporting families affected by dementia [37]. They step away from the medical model of nursing and follow a biopsychosocial approach that provides psychological and emotional support to families while at the same time helping them navigate the complexities of accessing health and social care support [37]. Although mainly community-based, in recent years, this model of dementia care has also shown promise when adapted to particular populations such as families affected by young onset dementia [38] and acute hospital settings [39]. Brown et al. [38] reported on family carers’ experiences of a specialist Lewy body dementia (LBD) Admiral Nurse service and concluded that, in line with what our study shows, the specialist knowledge and expertise of the LBD Admiral Nurses enabled family carers to better support the person with dementia and helped reduce the emotional stress associated with caring. Additionally, evidence of the cost-saving implications of an Admiral Nurse model is also starting to emerge. For example, a small pilot study [40] found that the Norfolk Admiral Nurse service meant a saving to the health and social care economy of GBP 426,601, hence demonstrating that it is possible to deliver gold-standard clinical care with no financial risk to the taxpayer. However, despite dementia being the leading cause of death among people in the UK, there are currently only 376 Admiral Nurses and only a very small proportion of these are available as formal support for families affected by YOD, which are more likely to have younger members [41]. This is in stark contrast to the more than 4000 Macmillan cancer nurses (specialist nurses who provide services for people living with cancer at every stage of the illness) [42], with which participants in our study drew parallels in terms of their vital role in supporting families [43]. Although still early days, we join other clinicians and researchers in advocating the need for Admiral Nurses-type service (delivered by nurses but potentially also by other relevant professionals) to be an essential, rather than an optional component of dementia care, not only in the UK but across the world. More specifically, our study provides evidence to support targeted research and clinical efforts to further adapt and tailor the service to reach and support families that include YDCs. We agree with Foster et al. [43] in calling for an increase in research efforts (targeting large-scale UK and international research funding) running alongside work on identifying effective implementation pathways to maximize the impact of research and its sustainability. We also encourage international efforts as we are confident that this whole-family approach model driven by Admiral Nurses is transferable and has the potential to be adapted to other health and social care systems across the world and to serve as a catalyst for the identification of YDCs.

To date, most of the research on YDCs (and young carers in general), including this study, has taken place mainly in a small pool of developed countries. Recent reviews looking at the experiences of families with young people affected by parental YOD [11,12,17] have included studies mainly from the UK, followed by the USA, Canada, and Australia. Very little is known about the situation of these families in developing countries. However, some studies are starting to emerge. Dourado et al. [44] compared YOD carers’ experiences in Brazil and Norway and reported that, regardless of cultural differences and disparities in access to support services, carers of people with YOD experienced highly similar challenges. We advocate for research and clinical efforts that consider knowledge across cultures, facilitating collaborative designs to deal with the shared challenges that YDCs and their families face.

### 4.1. Recommendations

Our findings support the need for a broad and holistic approach to the identification and support of YDCs. Although we see the identification of YDCs as the first step to supporting them directly, the support itself (if accessible, relatable, and tailored to their needs and their family’s needs) will play a role in improving subsequent identification. We recommend the following:Awareness building (involving active participation of young people) among professionals from adult and young people’s settings, young people themselves, and families. The focus should be on broadening the understanding of whole-family approaches and the concept of “caring” as a multifaceted role.Creation/improvement of communication pathways between adult and young people’s services (including young carers projects and schools). This responds to the realization that the issue of identification is interconnected and interdependent with both service settings.Adaptation of post-diagnosis assessments/visits by adult services professionals to include questions regarding younger family members and their potential involvement in supporting the person with dementia. To improve signposting and referral systems, we recommend the development of clear communication pathways across adult and young people’s organizations (including schools). Without addressing the communication gaps between young people’s services and adult services, the implementation of a whole-family approach will fail.Re-thinking and widening young carers charities’ marketing and engagement campaigns, challenging stigma, and promoting self-recognition, ensuring that the language and labels used resonate with YDCs. This can be achieved via active co-production of campaigns with YDCs. We recommend including case scenarios and examples of these particular cohorts in newsletters and campaigns and in tools already used within schools and other sectors.Developing peer support initiatives for this group, which might be delivered face to face or online. Although there are commonalities between the challenges linked to grandparental and parental dementia, services need to address the unique challenges and support these different situations. If not “fit for purpose”, they will inhibit the identification and subsequent engagement in support initiatives.Marketing and awareness campaigns to promote the role of Admiral Nurses and enable a more equitable distribution not only across the UK but also internationally. These campaigns should include clear evidence of the need for and value of this service. They should also include a step-by-step guide of the process that local authorities (in collaboration with third-sector organizations) need to follow to allocate the right resources for the setup and implementation of Admiral Nurse-type services. Uptake needs to go hand in hand with the generation of research evidence and the set up of effective ways for researchers to work with stakeholders to maximize the impact of such research.

### 4.2. Limitations

This study was able to recruit participants with relevant lived and professional experience. However, our sample only included one young adult carer (16+) and did not include dementia carers younger than 16. We acknowledge that these young people may have different reasons for not engaging in those articulated by their parents and professionals who took part. We are confident that this study provides detailed experiential information about the lived experiences and support structures of YDCs. However, further work, which involves active engagement of young carers from all sectors of society, including from all minority ethnic groups, will be required to refine, test, and implement our proposed recommendations. Finally, this study was conducted in the UK, a high-income country where young carers have specific legal rights and are represented in statutory guidance and national policy. We acknowledge this is a contextual limitation, but we believe that the work presented here is transferable and can be adapted to fit the needs of countries across the world who are at different stages in their journey to identify and support YDCs.

## 5. Conclusions

This study shows that YDCs are currently unsupported and unidentified by support services and the wider society. In response to this, we have explored identification pathways and have highlighted barriers and enablers to their implementation. The recommendations that we propose support the need for a broad and holistic approach that places YDCs and their families at the center of awareness-raising campaigns and service development efforts. We provide evidence that highlights the need to improve communication pathways between adult and children’s services. This responds to the realization that the issue of identification is interconnected and interdependent with both areas of practice and closely driven by a whole-family approach. Future research efforts should be focused on evaluating the multifaceted impact of a tailored YOD Admiral Nurse model. We believe that this model will increase the identification of YDCs and will also underpin the design of support services for these young people that are fit for purpose, which in turn will continue to improve identification.

## Figures and Tables

**Table 1 ijerph-20-07103-t001:** Summary information of study participants.

ID	Gender	Interview Completed	Attended Workshop	Category	Brief Description
**P1**	Female	Yes	Yes	Professional	Young Carers Strategy Lead for young carers project
**P2**	Female	Yes	Yes	Lived experience	Parent to two YDCs that helped care for their grandmother
**P3**	Female	Yes	Yes	Dementia Researcher	Area: stigma and perceptions of dementia among young people
**P4**	Female	Yes	Yes	Professional	Consultant Admiral Nurse
**P5**	Female	Yes	Yes	Professional	Admiral Nurse
**P6**	Male	Yes	Yes	Professional	Policy and Practice Manager for a carers (and young carers) charity
**P7**	Female	Yes	Yes	Lived experience	Parent to two YDCs that helped care for their father diagnosed with YOD
**P8**	Female	Yes	No	Lived experience	Parent to two YDCs that helped care for their grandfather
**P9**	Female	Yes	No	Professional	Council—Adult Services (wellbeing and activity programs)
**P10**	Female	Yes	No	Professional	Young Carer Service manager
**P11**	Female	Yes	Yes	Professional	YOD nurse specialist
**P12**	Female	Yes	Yes	Dementia Researcher/Lived experience	Area: impact of parental illness. Husband diagnosed with YOD when children were teenagers
**P13**	Female	Yes	Yes	Professional	Project Manager for a dementia support charity
**P14**	Female	Yes	Yes	Professional	Council—Age Friendly/Intergenerational coordinator
**P15**	Female	Yes	Yes	Lived experience	Parent to two YDCs that helped care for their father diagnosed with YOD
**P16**	Male	Yes	No	Dementia Researcher	Area: perceptions of dementia among young people, young people’s outcomes
**P17**	Male	Yes	Yes	Lived experience	Mother diagnosed with YOD at 49 (when he was 16)
**P18**	Female	No	Yes	Professional	Support Officer (young carers charity)
**P19**	Female	No	Yes	Professional	Council—Young carers officer

**Table 2 ijerph-20-07103-t002:** Identified themes and sub-themes.

Themes	Sub-Themes
**Theme 1:** A “whole-family approach”	Linking services and adapting processes Upskilling professionals (on family-centered approaches)
**Theme 2:** “Not a carer”—Self/family identification	This is our normal Reductionist definition of caring Issues around the label “young carer” Dealing with the “elephant in the room”
**Theme 3:** Postcode Lottery—high variability of support services	Admiral Nurses—the “gold standard” Schools—their role and level of engagement
**Theme 4:** Tailored support that is “fit for purpose”	
**Theme 5:** The “power” of peer support	

## Data Availability

If you wish to access the anonymized transcripts, please email the corresponding author.

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
