# Peer review of "Hard to Reach and Hidden: Improving the Identification of Young Dementia Carers"

_ijerph, 2023, doi:10.3390/ijerph20237103_

Round 1
Reviewer 1 Report
Comments and Suggestions for Authors
Thank you for the opportunity to review the manuscript. Overall, a current topic for a broader readership and further exploration of this topic is certainly unique, especially to explore the success (or failure) of YDCs identification pathways as well as the barriers and enablers to their implementation.
A few questions / comments and suggestions:
In line 86-94, consider providing a bit more detail on the sample, such as number of participants in each category (parents, professionals, etc.)
In line 117-124, specify that thematic analysis followed the phases outlined by Braun & Clarke, relevant to the study is not clear.
In line 95-115, specify any steps taken regarding research ethics and participant consent.
In line 137-406, provide a bit more interpretation and explanation of how the themes relate to the study aims and prior literature.
In line 504-506, “Little is known about these young people and their families, but this is particularly true for those living in developing countries. However, some studies are starting to emerge.”, relevant to the study is not clear.
The study limitations could be discussed in a bit more detail.
Comments on the Quality of English LanguageMinor editing of English language required.
Reviewer 2 Report
Comments and Suggestions for Authors
Thank-you for doing research on young carers, and the support that they require. I have some minor comments below, but overall think that the paper looks very good.
Line 32-33: In the UK, approximately 885,000 people are living with this condition.
Line 33: Please check the rules for writing numbers as numerals or words when there are several numbers in a sentence.
Line 126-127: Please check the following sentence – ‘Twenty-three potential participants were contacted through team members’ networks, none were recruited and none were recruited via social media.’
Line 153-155: Should the quote read ‘Where there’s dependents, …’?
Line 263: For non-UK readers, it may be worth explaining what a Macmillan nurse is.
Line 276-277: Please check the following sentence – ‘However, the pressures faced by schools a means that, 276 according to participants, …’
Line 524: spelling error – currently reads inclusing rather than including.
Have you considered adding what the strengths and limitations of the research are. For example:
1) I note that you only included young carers age 16+ (which I am guessing due to consent reasons) but the younger group may have different reasons for not engaging.
2) I note you only have 1 participant that is a young carer themselves, and young carers may have other reasons not already noted why they don’t engage.
Comments on the Quality of English LanguageMinor spelling errors as noted above.
